# FLock: Robust and Privacy-Preserving Federated Learning based on Practical Blockchain State Channels

Submission Id: 1878

## Abstract

*Federated Learning* (FL) is a distributed machine learning paradigm that allows multiple clients to train models collaboratively without sharing local data. Numerous works have explored security and privacy protection in FL, as well as its integration with blockchain technology. However, existing FL works still face critical issues. i) It is difficult to achieving *poisoning robustness* and *data privacy* while ensuring high *model accuracy*. Malicious clients can launch *poisoning attacks* that degrade the global model. Besides, aggregators can infer private data from the gradients, causing *privacy leakages*. Existing privacy-preserving poisoning defense FL solutions suffer from decreased model accuracy and high computational overhead. ii) Blockchain-assisted FL records iterative gradient updates on-chain to prevent model tampering, yet existing schemes are not compatible with practical blockchains and incur high costs for maintaining the gradients on-chain. Besides, incentives are overlooked, where unfair reward distribution hinders the sustainable development of the FL community. In this work, we propose FLock, a robust and privacy-preserving FL scheme based on practical blockchain state channels. First, we propose a lightweight secure *Multi-party Computation* (MPC)-friendly robust aggregation method through quantization, median, and Hamming distance, which could resist poisoning attacks against up to $< 50\%$ malicious clients. Besides, we propose communication-efficient Shamir's secret sharing-based MPC protocols to protect data privacy with high model accuracy. Second, we utilize blockchain off-chain state channels to achieve immutable model records and incentive distribution. FLock achieves cost-effective compatibility with practical cryptocurrency platforms, *e.g.* Ethereum, along with fair incentives, by merging the secure aggregation into a multi-party state channel. In addition, a pipelined *Byzantine Fault-Tolerant* (BFT) consensus is integrated where each aggregator can reconstruct the final aggregated results. Lastly, we implement FLock and the evaluation results demonstrate that FLock enhances robustness and privacy, while maintaining efficiency and high model accuracy. Even with 25 aggregators and 100 clients, FLock can complete one secure aggregation for ResNet in 2 minutes over a WAN. FLock successfully implements secure aggregation with such a large number of aggregators, thereby enhancing the fault tolerance of the aggregation.

## CCS Concepts

• **Security and privacy** → *Distributed systems security*.

## 1 Introduction

As a distributed machine learning paradigm, *Federated Learning* (FL) [34, 41, 63] has achieved rapid development in recent years. FL allows multiple clients to train a model collaboratively without sharing local data. Each client uses its locally kept dataset to train the model and then sends the gradients to the aggregator, which aggregates all received gradients to update the model. Through multiple rounds of updates, the global model is gradually optimized. To avoid centralization failure, decentralized FL has been widely studied [36, 53], which aims to eliminate dependence on the centralized server. In decentralized FL, multiple aggregators participate in model aggregation or clients communicate and collaborate directly through a *Peer-to-Peer* (P2P) network to achieve model training and updates, which greatly enhances the reliability of the FL system.

FL is developing rapidly and has been used in highly sensitive areas, *e.g.*, medical [1, 12] and financial [68]. Moreover, its combination with Web 3.0 [26, 59] provides innovative solutions for data rights confirmation, transparency, and incentive mechanisms. As the next generation of the Internet, Web 3.0, based on blockchain technology, realizes decentralization, trustlessness, and user data rights confirmation, which is a natural fit with the distribution and privacy protection of FL. Currently, FL is in its early stages and still faces serious issues in security and practicability.

**Issue-❶ FL robustness and privacy with high accuracy.** In FL, achieving *poisoning robustness* and *data privacy* while ensuring high *model accuracy* is a critical issue that requires urgent attention.

i) *Poisoning robustness*: Clients in FL are highly distributed, so it is hard to guarantee their behaviors. Malicious clients might intentionally manipulate the global model by submitting tampered or falsified gradients, undermining the FL system's overall effectiveness [21, 55]. To address this problem, existing works proposed various poisoning robustness methods to resist malicious clients [8, 11, 13, 24, 46, 62, 64, 67]. These approaches use a variety of similarity metrics (*e.g.*, Euclidean norm [8] and cosine distance [11]) and advanced detection algorithms [8, 11, 46] to discover the abnormal gradients and mitigate potential threats. However, implementing poisoning defenses on privacy-protected data is more challenging, and imprecise defense mechanisms may filter out honest clients' gradients, leading to a decline in model accuracy.

ii) *Data privacy*: When the gradients are in plaintext, the aggregators can infer private client data or data property from the gradients. Existing works have developed different secure aggregation schemes based on *Secure Multi-party Computation* (MPC) [14, 19, 23, 49], masking [6, 9, 39], *homomorphic encryption* (HE) [3, 65], functional encryption [40] and *differential privacy* (DP) [45, 51]. However, most of them focus on simple SUM/Average aggregation, which is vulnerable to poisoning attacks.

To ensure both poisoning robustness and data privacy, trivially combining existing robust aggregation methods with MPC techniques presents several issues. Existing robust aggregation methods mostly rely on complex operations (*e.g.*, cosine similarity) that are MPC-unfriendly with decreased model accuracy, while causing high computation overhead [46]. While integrating MPC with customized lightweight robust [19, 23, 49] to defend against both attacks are restricted to 2- or 3-party settings (*a.k.a.*, two or three non-colluding aggregators). This setting is difficult to extend to

practical distributed environments with more aggregators, and it fails to achieve fault tolerance for the aggregators.

**Issue-❷ Practical compatibility to existing blockchains.** The structure of FL is distributed and highly consistent with the architecture of the blockchain. As a distributed ledger, blockchain enables decentralized data management through its chain structure and consensus mechanisms, which provide a potential solution for decentralized FL. The gradients of optimization iterations during training could be uploaded on the chain, which prevents model tampering and maintains the current global model for all participants. However, there exist several problems:

i) Most blockchain-based FL schemes utilize specialized consensus better suited for consortium chains than widely adopted platforms like Bitcoin [43] or Ethereum [60], which restricts the real-world application and poses significant challenges to deployment and scalability. Besides, to fully satisfy the decentralization requirements of FL, several works have attempted to implement vanilla FL (non-robustness or non-privacy protection) on top of the blockchain [2, 66]. Moreover, the aggregation process is highly dependent on the centralized aggregation server. Although some existing works attempt to construct FL frameworks based on blockchain, many still depend on a single server for aggregation [42, 57], which contradicts the high decentralization of the blockchain.

ii) Unfair incentives, or a lack thereof, may demotivate clients with greater contributions in FL, adversely affecting the performance of the global model. Furthermore, these imbalances can lead some participants to reduce their contributions or even upload low-quality local updates while still reaping benefits from the global model. To address these issues, the incentive mechanisms in FL must prioritize fairness, ensuring that the contributions of each participant are reasonably recognized and rewarded.

Based on the above analyses, a burning question arises:

*Can we design a FL scheme that not only achieves poisoning robustness, data privacy, and high model accuracy but also integrates into practical blockchains at low cost with fair incentives?*

In this work, we propose FLock, a robust and privacy-preserving FL framework based on practical blockchain state channels, to answer this question affirmatively. In FLock, we leverage the quantization, Hamming distance optimizations, and median computation to resist the poisoning attacks against malicious training clients and make use of Shamir's secret sharing (SS)-based MPC protocol to resist the inference attacks. Moreover, FLock takes advantage of blockchain state channels to achieve low overhead decentralized FL with fair incentives. To our knowledge, FLock is the first to achieve low-cost decentralized FL by using state channels.

**Contributions.** Our main contributions are summarized as follows:

• **Lightweight MPC-friendly robust aggregation scheme.** To defend against poisoning attacks from malicious clients, we propose a lightweight, MPC-friendly robust aggregation method that leverages quantization and Hamming distance optimizations, our approach operates in an honest-majority setting (up to 50% malicious clients) without needing a root-dataset. We propose Shamir's SS-based MPC protocols for securely evaluating our aggregation method with high model accuracy. Our protocols can be deployed on arbitrary aggregators and tolerate a certain number of crashes. Additionally, we introduce several novel optimizations that significantly reduce communication costs.

• **Compatibility with practical off-chain channels blockchain platforms and fair incentives.** FLock achieves low-cost compatibility with practical cryptocurrencies, such as Ethereum, by putting the entire aggregation process in a multi-party state channel with a pipelined *Byzantine Fault-Tolerant* (BFT) consensus. To our knowledge, FLock is the first to combine blockchain off-chain state channels and FL to realize low on-chain cost while ensuring model immutability. Besides, FLock introduces a fair incentive distribution through smart contracts according to the contributions of aggregators and clients.

• **Implementation and evaluation.** We implement FLock and compare the poisoning tolerance and aggregation efficiency with existing protocols. Besides, we evaluate the on-chain and off-chain overhead involved in FLock. The evaluation results show the robustness of this work and demonstrate that even though our work enhances privacy and security, it remains efficient. Even with 25 aggregators and 100 clients, FLock can complete a secure aggregation for ResNet in less than 120 seconds with around 5GB communication size. There is currently little work evaluating the performance of 25 aggregators. Besides, the on-chain and off-chain overheads show the practicality of FLock.

## 2 Preliminaries

### 2.1 Notations

Let $C_l$ denote the $l$-th training client and $l \in [m]$. $P_j$ denote the $j$-th aggregator and $j \in [n]$. $\boldsymbol{g}$ denotes the gradient vector of $(\boldsymbol{g}_1, \ldots, \boldsymbol{g}_K)$ of size $K$. $\boldsymbol{g}_k^{(l)}$ is the $k$-th gradient component from the $l$-th training client. By default, $\overline{\boldsymbol{g}}$ is the origin gradient before quantization, $\widehat{\boldsymbol{g}}$ indicates the sign-quantized gradient $\widetilde{\boldsymbol{g}}$ with values in $\{-1, +1\}$, and $\boldsymbol{g}$ denotes Boolean encodings (*a.k.a.*, values in $\{0, 1\}$) of $\widehat{\boldsymbol{g}}$. And $\langle \cdot \rangle$ is utilized for SS.

### 2.2 Federated Learning

FL is a decentralized machine learning approach where clients collaboratively train a global model under the coordination of aggregators, utilizing local data without sharing the original datasets. We focus on horizontal scenarios where data follows an independent and identically distributed (*i.i.d.*) pattern and optimize the model using *Stochastic Gradient Descent* (SGD) [70] in this work. The main process of FL is as follows.

• Local training: Each client downloads the initial global model from the server, and performs multiple training iterations locally without sharing data to generate gradient updates.

• Model aggregation: Each client sends its updated gradients to the server. The server aggregates the gradients from all clients to generate the global model update.

• Global model updating: Once the gradients from all clients are aggregated, the server updates the global model and distributes it to each client for the next round of iterative local training.

Additionally, FL's security issues cannot be ignored. *Poisoning attack* can affect the training results of the global model by submitting malicious model updates, and even cause the global model to deviate. Besides, the server may launch *inference attacks* to infer the local private data or sensitive information of the clients from gradients.

**Table 1: Comparison with state-of-the-art FL frameworks.**

| | Security | | Decentralization | | # of Aggregators | |
|---|---|---|---|---|---|---|
| | Data Privacy | Poisoning Robustness | Blockchain-based | Cryptocurrency Compatibility | Single/Two-party | Multiple |
| Krum [8] | - | Euclidean norm | ○ | ○ | ● | ○ |
| [64] | - | Trim-mean + median | ○ | ○ | ● | ○ |
| FLTrust [11] | - | Root-dataset + cosine similarity | ○ | ○ | ● | ○ |
| ELSA [49] | Boolean SS | Euclidean norm | ○ | ○ | ● | ○ |
| FLAME [45] | DP | Euclidean norm | ○ | ○ | ● | ○ |
| RoFL [37] | Masking | Euclidean norm | ○ | ○ | ● | ○ |
| [6, 9, 39] | Masking | - | ○ | ○ | ● | ○ |
| FLOD [19] | SS + AHE | Root-dataset + Hamming distance + quantization | ○ | ○ | ● | ○ |
| TGFL [66] | - | Data poisoning resistance | ● | ○ | ○ | ● |
| BDVFL [57] | Masking | - | ● | ○ | ● | ○ |
| Biscotti [51] | DP | Krum | ● | ○ | ○ | ● |
| PBFL [42] | FHE | Cosine similarity | ● | ○ | ● | ○ |
| BlockDFL [48] | Gradient compression | Median-based testing + Krum | ● | ○ | ○ | ● |
| FLock | Shamir SS-based MPC | Quantization + median + Hamming distance | ● | ● | ○ | ● |

○ Not Support ● Support

## 2.3 Building Blocks Related to Blockchain

Blockchain is a distributed ledger that achieves transparency, immutability, and security of information by storing data records on all blockchain network nodes. Smart contracts are self-executing codes deployed on the blockchain that can be automatically triggered based on preset conditions. The logic of smart contracts runs through the blockchain network, which is transparent and tamper-proof, reducing manual intervention and trust assumptions.

### 2.3.1 Multi-party State Channels.
To address blockchain scalability issues, state channels offer an off-chain solution that facilitates numerous transactions and states updating off-chain, with only the channel create and close recorded on-chain. Moreover, multi-party state channels enable multiple participants to interact within a single off-chain channel, allowing any participant to update the channel while all participants collaboratively maintain its state. Assuming there are $n$ parties involved in a channel. The process of a multi-party state channel is outlined as follows.

- **Create:** Each party $P_i$ intending to participate in the channel deploys a state channel contract and deposits their respective initial amounts as their initial state $st_{P_i}$. The initial channel state is $ST_0 = \{st_{P_i}\}_{1 \le i \le n}$.
- **Update:** Any participant in the channel can initiate a state update request, which will be broadcast to all participants within the channel. The state update will be valid after all (or threshold) participants agree, and all participants will jointly maintain the latest state of the channel.
- **Close:** If a participant initiates a closing request and all participants agree on the final state of the channel, the channel can be closed with the final state, and the close transaction will be recorded on the chain. Otherwise, others can raise a dispute with the latest state, and the invalid request will be rejected.

### 2.3.2 Pipelined Multi-signature BFT Consensus.
To formally describe the secure aggregation of participants within the channel, we leverage a pipelined multi-signature BFT consensus, PMSBFT. The pipelining setting allows proposals from different phases to be processed in different consensus phases at the same time, which enables the model gradients for different tasks to be aggregated and processed in parallel, improving the efficiency and practicality of model aggregation. PMSBFT adopts a stable leader, ensuring an efficient and stable process when the leader behaves honestly. The concrete PMSBFT protocol is described in Appendix A.

## 2.4 Shamir's Secret Sharing-based MPC

Shamir's SS is widely used to construct efficient MPC protocols [17, 35], and consists of five basic subprotocols:

- SS.Setup($1^\kappa, n$) → $(pp, \{vpk_i, vsk_i\}_{i \in [n]})$: Initialize the public parameter $pp$. Let $\mathbb{F}_p$ be a finite field modulo $p$. Generate the public-private key pair $(vpk_i, vsk_i)$ for each participant $P_i$ for $1 \le i \le n$.
- SS.Share($\{s_k\}_{k \in [K]}$) → $(\{C_k\}_{k \in [K]}, \{s_{ik}\}_{k \in [K], i \in [n]})$: For each secret $s_k$, dealer randomly selects a degree-$t$ polynomial $F(\cdot, k) \in \mathbb{F}_p[\cdot]$, s.t. $F(0, k) = s_k$. Compute $s_{ik} = F(i, k)$ as the share of $s_k$ to $P_i$. $\langle s_k \rangle^t$ denotes the shares of $(s_{1k}, \ldots, s_{nk})$ with degree $t$ and $\langle s_k \rangle$ indicates the degree $t$ by default. Each party $P_i$ receives $K$ polynomial shares.
- SS.Reconstruct($\{s_{ik}\}_{k \in [K], |s_{ik}| \ge t}$) → $(\{s_k\}_{k \in [K]})$: Take the secret shares $\{s_{ik}\}_{k \in [K]}$ as inputs, and output the original secrets $\{s_k\}_{k \in [K]}$ if the number of valid shares $|s_{ik}| \ge t$.
- SS.Addition($\langle a \rangle, \langle b \rangle$) → $\langle a + b \rangle$: Let $\langle a \rangle$ and $\langle b \rangle$ denote two shares of secrets $a, b \in \mathbb{F}_p$, each party locally adds its secret shares to get $\langle a + b \rangle$.
- SS.Multiplication: Let $\langle a \rangle$ and $\langle b \rangle$ be two degree-$t$ shared secret inputs, the parties can compute degree-$2t$ intermediate $\langle z \rangle^{2t}$ by computing their share locally. With the double sharing $(\langle r \rangle, \langle r \rangle^{2t})$ of a random $r \xleftarrow{\$} \mathbb{F}$, the parties can reduce the degree of $\langle z \rangle^{2t}$ to $t$ as follows: i) all parties locally compute $\langle c \rangle^{2t} = \langle z \rangle^{2t} + \langle r \rangle^{2t}$, ii) collaboratively reconstruct $c$, and iii) finally compute $\langle a \cdot b \rangle = c - \langle r \rangle$. And $(\langle r \rangle, \langle r \rangle^{2t})$ can be generated using the randomness extraction method [17].

On top of them, existing works have built various fast protocols for more complex functions, including less-than (<, LT). We use protocol LT of [35] for secure comparison in a black-box manner. For convenience, we utilize + and · to represent SS.Addition and SS.Multiplication implicitly in this work. Also, the above protocols can be easily extended to vectors and matrices [35] in parallel, we also use these technologies in our protocols.

## 3 System Design

We capture the communication model, threat model, and security goals in this section.

### 3.1 Communication Model

In this work, we consider the synchronous model. Assuming that all communication processes are composed of rounds. The size of a round can be adjusted according to the actual situation, which

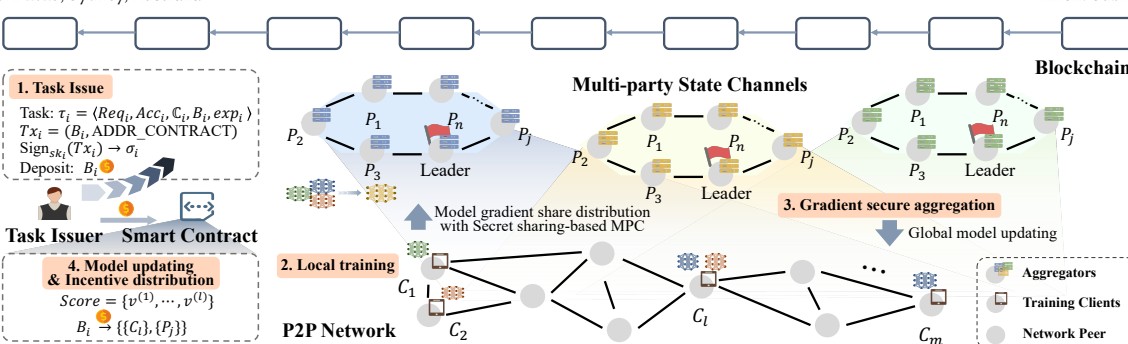

**Figure 1: The workflow of the protocol**

could be 1 second in the real world. Let $\Delta$ be the upper bound of time delay. In the update phase within the multi-party channels, to ensure that gradient shares can be reconstructed and channel state updates can be processed quickly, the protocol adopts BFT consensus with partial synchronization, which can also run under a synchronous network.

### 3.2 Threat Model

We capture two kinds of adversaries in this work: i) We assume that all participant aggregators in the multi-party state channel are semi-honest, which means they will not deviate from the predetermined process of the protocol, but will try to obtain as much private information as possible during the execution. However, the MPC protocol does not protect data before entering the channel. ii) In training clients, we assume that the malicious adversary $\mathcal{A}$ can control no more than 1/3 of any participants at the beginning of the protocol and access all information of the corrupted participants. There are $n \geq 3f + 1$ participants within a channel, where $f$ is the maximum number of corrupted participants. The training clients could be seen as malicious and may launch poisoning attacks.

### 3.3 Security Goals

In this work, we consider the following two security goals.
**(G1) Poisoning robustness.** When the FL system is attacked by malicious clients, it can still maintain the accuracy, unbiasedness, and fairness of the model to a certain extent.
**(G2) Cryptographic privacy.** Ensure that the aggregators cannot infer sensitive information about the training data by accessing the model or gradients.

### 4 Concrete Protocol

In this section, we first give an overview and then present FLock in detail. The workflow is illustrated in Figure 1.

### 4.1 Overview

**Task issue.** The task issuer $I_i$ publishes its task through a smart contract. Then $I_i$ pays a deposit to the contract address, which is used as a reward for the training clients and the aggregation participants in the channel. When the contract result is on the chain, the task will be added to a pending set, and all peers can choose whether to participate in training or gradient aggregation.
**Local training.** All peers who are willing to participate in the training obtain the model weights and train the model to generate updated model gradients. Then the training clients $C_l$ $(1 \leq l \leq m)$

quantize the updated gradient components into the Boolean format with SIGNSGD [7] and the SIGNSGD $\rightarrow$ Boolean conversion for the subsequent calculation of Hamming distance in aggregation. Each training client $C_l$ generates $n$ shares of the quantified gradient components through SS.Share($\cdot$) and distributes the shares to the participants in the multi-party state channel for secure aggregation.
**Gradient secure aggregation.** The peers who are willing to participate in gradient aggregation for a task create a multi-party state channel and wait for gradient shares from training clients. The peers participating in the aggregation execute the process through the off-chain state channels, making the overall gradient aggregation process compatible with existing cryptocurrency platforms and making the protocol more practical. Each participant $P_j$ first verifies the validity of the shares by SS.Verify($\cdot$). Then $P_j$ calculates the Hamming distance of the shares received from each training client $C_l$ to obtain a score $v^{(l)}$ and then performs a weighted average of the shares according to the scores for gradient share aggregation. The participants then reconstruct and synchronize the aggregated gradient through the pipelined multi-signature BFT consensus, PMSBFT, within the channel with SS.Reconstruct($\cdot$).
**Model updating and incentive distribution.** Finally, the channel leader $\mathcal{L}$ returns the aggregated gradients to clients for iterative training. If the current model has satisfied the requirements of the task issuer, $\mathcal{L}$ submits the current model to the smart contract, which will pay the participants $\{P_j\}_{j \in [n]}$ in the channel and pay the training clients $\{C_l\}_{l \in [m]}$ according to the scores $\{v^{(l)}\}_{l \in [m]}$.

For the sake of clarity, we introduce the plaintext training workflow in § 4.2, add the MPC-based privacy protection in § 4.3, and describe how to be compatible with practical blockchain by state channels and provide fair incentives in § 4.4.

### 4.2 Plaintext Training Workflow

Our training method is inspired by FLOD [19]. Unlike FLOD which requires a clean root-dataset to bootstrap trust against a majority of malicious clients, our approach operates in an honest-majority setting (up to 50% malicious clients) without needing a root-dataset. This trade-off is practical, as the honest-majority assumption is widely acknowledged while obtaining a clean root-dataset is often challenging in decentralized settings.
**[Local Training]** After each client $C_l$ $(1 \leq l \leq m)$ completes training locally, the original floating gradients provide a wider space for possible attacks. Then $C_l$ transforms each gradient component into $\{-1, 1\}$-encodings (denoted as $\widehat{g}^{(l)} = (\widehat{g}_1^{(l)}, \ldots, \widehat{g}_K^{(l)})$) as SIGNSGD [7]. This conversion could effectively limit the attack

space of the adversary, where the gradients are restricted to discrete values. Then, $C_l$ further converts $\widehat{g}^{(l)}$ into Boolean type through the following conversion process:

SIGNSGD→Boolean conversion: Given $\widehat{g}^{(l)}$, for each element $\widehat{g}_k^{(l)}$ with $1 \le k \le K$, we have

$$g_k^{(l)} = \begin{cases} 0, & \text{if } \widehat{g}_k^{(l)} = 1, \\ 1, & \text{otherwise.} \end{cases} \tag{1}$$

All the above computations are executed locally on the training client. Then, $C_l$ obtains the Boolean gradients $g^{(l)}$ send them to aggregator $P$. Note we only send the Boolean gradients without $\{-1, 1\}$-encoded ones. Looking ahead, this is because if sends both, we will need to ensure consistency[1] of them in secure aggregation, which is expensive in MPC.

**[Gradient aggregation]** After receiving the gradients $\{g^{(l)}\}_{l \in [m]}$, aggregator $P$ first converts $g^{(l)}$ into $\{-1 + 1\}$-encodings for the subsequent aggregation by

$$\widehat{g}^{(l)} = 1 - 2 \cdot g^{(l)}. \tag{2}$$

Then, $P_j$ sums $\{\widehat{g}^{(l)}\}_{l \in [m]}$ in element-wise, i.e., $\widehat{g}_k^{(s)} = \sum_{l=1}^m \widehat{g}_k^{(l)}$. Next, $P$ computes $g^{(k)}$ to bootstrap trust in honest-majority as

$$g_k^{(s)} = \begin{cases} 0, & \text{if } \widehat{g}_k^{(s)} \ge 0, \\ 1, & \text{otherwise.} \end{cases} \tag{3}$$

It is easy to see $g^{(s)}$ excludes 50% of the malicious clients in element-wise since it is determined by major gradients' values. Although $g^{(s)}$ can also represent honest aggregated results, too much information is lost, leading to accuracy degradation of the global model. To keep as many honest gradients as possible, we leverage the Hamming distance-based weighted aggregation [19].

Hamming distance computing: Aggregator $P$ computes the Hamming distance between $g^{(s)}$ and $g^{(l)}$ from each $C_l$ as

$$hd^{(l)} = \sum_{k=1}^K (g_k^{(l)} \oplus g_k^{(s)}). \tag{4}$$

Then, we compute a score for each client based on $hd^{(l)}$. Intuitionally, a smaller Hamming distance means that the gradient is closer to $g^{(s)}$, a.k.a., more likely to be sent by an honest client, and thus should have a higher score. The score is calculated as follows.

$\lambda$-Score computing: Compute the score $v^{(l)}$ for each client $C_l$ as

$$v^{(l)} = \begin{cases} \lambda - hd^{(l)}, & \text{if } hd^{(l)} < \lambda \\ 0, & \text{otherwise} \end{cases} \tag{5}$$

Weighted aggregation: Aggregator $P$ weighted aggregates the gradients as:

$$g = \sum_{l=1}^m \frac{v^{(l)}}{\sum_{\tau=1}^m v^{(\tau)}} \widehat{g}^{(l)} \tag{6}$$

Noted that in the weighted aggregation, we use the $\{-1, +1\}$-encoded gradients following existing works [7, 19].

**Security & privacy concerns.** During the above aggregation method, $\widehat{g}^{(s)}$ and Hamming distance-based score could provide resistance to poisoning attacks, which excludes $< 50\%$ malicious

---

[1] Boolean and $\{-1, 1\}$-encoded gradients satisfy equation (1)

clients. Unfortunately, it does not consider the resistance to privacy attacks, where the aggregators could infer the private information of the local dataset from gradients sent by training clients.

## 4.3 MPC-based Secure Aggregation

We leverage Shamir's SS to design MPC protocols for securely evaluating our aggregation method. Unlike previous approaches limited to 2 or 3 aggregators [19, 23, 49], our secure aggregation can be deployed on arbitrary aggregators and tolerates a certain number of crashes benefiting from Shamir's SS. Additionally, we introduce several novel optimizations that significantly reduce communication costs, which may be of independent interest.

**[Gradient secure aggregation]** To protect data privacy, we design secret sharing-based MPC protocols to achieve secure aggregation for the plaintext training procedure in § 4.2. At a high level, we employ several aggregators $\{P_j\}_{j=1}^n$ and develop four subprotocols: $\Pi_{\text{Boostrap}}, \Pi_{\text{HM}}, \Pi_{\lambda\text{Score}}$, and $\Pi_{\text{WA}}$, for each step of the aggregation.

### 4.3.1 Protocol $\Pi_{\text{Boostrap}}$. 
After receiving its gradients shares $\langle g^{(l)} \rangle_j$ from client $C_l$, $P_j$ first need to check *whether every element of $g^{(l)}$ lies in $\{0, 1\}$ or not*. This is trivial in plaintext but challenging in MPC since aggregators cannot access the true values.

Given $\forall x$, we observe that $x \in \{0, 1\} \Leftrightarrow x \cdot (1 - x) = 0$. Similarly, for each $\langle g \rangle^{(l)}$, all aggregators can also collaboratively compute

$$\langle c^{(l)} \rangle = \langle g^{(l)} \rangle \cdot (1 - \langle g^{(l)} \rangle), \tag{7}$$

open $c^{(l)}$, and check whether $c^{(l)}$ is all 0 or not. However, this requires a communication of $O(mKt)$, which is dependent on $K$. To reduce the communication, we propose two optimizations:

(1) **Probabilistic Test.** After getting $c^{(l)}$, we interpret vector $c^{(l)} = [c_0^{(l)}, c_2^{(l)}, \ldots, c_{K-1}^{(l)}]$ as $K$ coefficients of a degree-$(K-1)$ polynomial. Then, we can use the Schwartz–Zippel lemma [50, 71] for the polynomial identity test. Roughly, we let all parties sample a common random $r \xleftarrow{\$} \mathbb{F}$, compute $\sigma^{(l)} = \sum_{i=0}^{K-1} c_i^{(l)} r^i$, and check $\sigma^{(l)} = 0$ or not.

(2) **Test-then-Open.** Recall that all values are in the secret-shared fashion, if we follow existing secure multiplication, the parties need to reduce the degree for the whole vector $c^{(l)}$ in equation (7) before polynomial identity test. However, we can post the degree reduction after the polynomials computation: assuming $\langle g^{(l)} \rangle$ is of degree-$t$, then all aggregators first compute degree-$2t$ $\langle c^{(l)} \rangle^{2t}$ locally. Next, we let all aggregators locally compute $\langle \sigma^{(l)} \rangle^{2t} = \sum_{i=0}^{n-1} \langle c_i^{(l)} \rangle^{2t} r^i$. Finally, all aggregators only need to open one value $\sigma^{(l)}$, instead of the vector $c^{(l)}$.

With the above two optimizations, we only need $O(mt)$ communication for checking whether $g^{(l)} \in \{0, 1\}^K$ for $l \in [m]$. Also, with the Schwartz–Zippel lemma, when $\sigma^{(l)} = 0$, we have at least $1 - \frac{K-1}{|\mathbb{F}|}$ probability to guarantee that $c^{(l)}$ is composed of all 0. For widely used neural networks with millions of parameters (a.k.a., $n < 2^{16}$), we can choose 60 bits finite field $\mathbb{F}$, and we have probability at least $1 - 2^{-40}$ guarantee probabilistic test works. After the above check, the aggregators discard gradients with $\sigma \ne 0$.

Below, all aggregators process the qualified $\langle g^{(l)} \rangle$ to get $\langle g^{(s)} \rangle$: i) each $P_j$ locally convert its $\langle g^{(l)} \rangle$ from Boolean encoding to $\{-1, +1\}$-encoding as $\langle \widehat{g}^{(l)} \rangle = 1 - 2\langle g^{(l)} \rangle$. ii) each $P_j$ sums its

shares in element-wise to obtain $\langle \widehat{g}^{(s)} \rangle = \sum_{l=1}^{m} \langle \widehat{g}^{(l)} \rangle$, and collaboratively computes $\langle g^{(s)} \rangle = 1 - (\langle \widehat{g}^{(s)} \rangle \geq 0)$, where the comparison $(\langle \widehat{g}^{(s)} \rangle \geq 0)$ can be implemented by Less-Than protocol LT of [35].

*4.3.2 Protocol* $\Pi_{\mathsf{HM}}$. In this protocol, we securely compute the Hamming distance between $\langle g^{(l)} \rangle$ and $\langle g^{(s)} \rangle$ for $l \in [K]$. Since the gradients are secretly shared in Shamir secret sharing over field $\mathbb{F}$, we need to compute the Hamming distance using *arithmetic* operations over $\mathbb{F}$. One trivial solution is as follows: first, we compute the element-wise XOR as:

$$\langle d^{(l)} \rangle = \langle g^{(l)} \rangle + \langle g^{(s)} \rangle - 2 \cdot \langle g^{(l)} \rangle \cdot \langle g^{(s)} \rangle. \qquad (8)$$

Then, $P_j$ can locally compute

$$\langle hd^{(l)} \rangle = \sum_{k=0}^{K-1} \langle d_k^{(l)} \rangle. \qquad (9)$$

However, this method requires communicating $O(mKt)$ field elements for the secure multiplication in equation (8). To reduce the communication costs, we exploit *Sum-then-DegReduce* technique: In the secure multiplication $\langle g^{(l)} \rangle \cdot \langle g^{(s)} \rangle$, after getting $\langle g^{(l)} \cdot g^{(s)} \rangle^{2t}$, instead of conducting degree-reduction immediately, we can first perform summation. Concretely, we interpret $\langle g^{(l)} \rangle$ and $\langle g^{(s)} \rangle$ as degree-$2t$ Shamir secret shares (by padding the coefficients of degree-$t$ to dgree-$(2t-1)$ as 0), then we compute $\langle d^{(l)} \rangle^{2t}$. Next, we compute $\langle hd^{(l)} \rangle^{2t}$ similar as equation (9) but with degree-$2t$ sharing. Finally, we only need to reduce the degree of $\langle hd^{(l)} \rangle^{2t}$ to $t$. In this way, our total communication for computing Hamming distance is reduced to $O(mt)$, not dependent on $K$.

*4.3.3 Protocol* $\Pi_{\lambda \mathsf{Score}}$. With the Hamming distance $\langle hd^{(l)} \rangle$, $P_j$ computes the score $\langle v^{(l)} \rangle$ for the gradients of $C_l$ as:

$$\langle v^{(l)} \rangle = (\lambda - hd^{(l)}) \cdot ((\lambda - \langle hd^{(l)} \rangle) \geq 0), \qquad (10)$$

where the comparison $((\lambda - \langle hd^{(l)} \rangle) \geq 0)$ can be computed using protocol LT. In this way, when $(\lambda - hd^{(l)}) \geq 0 \Leftrightarrow hd^{(l)} \leq \lambda$, we get $v^{(l)} = \lambda - hd^{(l)}$; otherwise, we set $v^{(l)} = 0$, computing $\lambda$-Score securely in MPC.

*4.3.4 Protocol* $\Pi_{\mathsf{WA}}$. Simply computing equation (6) involves expensive secure division, we thus decompose it as follows:

1) We compute $\langle \mathcal{V} \rangle = \sum_{l=1}^{m} \langle \langle v \rangle^{(l)} \rangle$ and reveal $\mathcal{V}$.
2) The aggregated results can be computed as $\langle g \rangle = \sum_{l=1}^{m} \frac{1}{\mathcal{V}} \cdot (\langle v^{(l)} \rangle \cdot \langle g^{(l)} \rangle)$, where $\frac{1}{\mathcal{V}}$ is computed in plaintext, and we only need secure addition and multiplication.

However, the above procedure still requires a communication of $O(mKt)$ field elements. Hence, we use *Sum-then-DegReduce* to reduce the communication. In detail, we first get $\langle v^{(l)} \cdot g^{(l)} \rangle^{2t}$ when computing $(\langle v^{(l)} \rangle \cdot \langle g^{(l)} \rangle)$, summing up to get $\langle g \rangle^{2t} = \sum_{l=1}^{m} \frac{1}{\mathcal{V}} \cdot \langle v^{(l)} \cdot g^{(l)} \rangle^{2t}$, and finally reduce the degree of $\langle g \rangle^{2t}$ to $t$. In this way, we only require a communication of $O(Kt)$, achieving a reduction of $m \times$. Although we reveal $\mathcal{V}$ in step 1), it saves the expensive costs of secure division. And revealing $\mathcal{V}$ alone will not leak the values or distributions of gradients. We think it is a reasonable trade-off between efficiency and security.

By integrating all the subprotocols outlined in § 4.3.1-§ 4.3.4, we construct our secure $n$-party aggregation for robust FL.

## 4.4 Blockchain Compatibility & Fair Incentive

The above process only involves a privacy-preserving FL training scheme but does not mention compatibility with practical blockchain and incentive distribution. Then, we introduce how to merge the above FL scheme into practical blockchain state channels with fair incentive distribution.

**[Task issue]** First, to add an incentive mechanism, we introduce the task issue process, where the task issuer deposits the rewards for the training clients and aggregators. We leverage smart contracts to achieve fair incentives, where the task issuer cannot deny the published tasks and the promised rewards. The task issue process can be found in Appendix B.

**[Gradient secure aggregation]** We enable gradient secure aggregation off-chain by leveraging multi-party state channels with the pipelined multi-signature BFT consensus, PMSBFT, improving compatibility with most cryptocurrency platforms. To verify the validity of shares received from clients, we consider introducing the verifiability of SS. Besides, we leverage *batching* to optimize efficiency, which allows the dealer to share multiple secrets in parallel and aggregators can verify the validity of shares at one time. Specifically, introduce a polynomial commitment PolyCommit$(\cdot)$ in SS.Share and generate a commitment $C_k \leftarrow \mathsf{PolyCommit}(F(\cdot, k))$ for $k \in [K]$. Then, adding a verification algorithm SS.Verify$(\{C_k\}_{k \in [K]}, \{s_{ik}\}_{k \in [K], i \in [n]}) \rightarrow b$, which outputs a indicator bit $b \in \{0, 1\}$ to indicate whether the secret shares $\{C_k\}_{k \in [K]}$ are valid.

**Aggregated gradient reconstruction within the channel.** All peers who are willing to join the model aggregation as aggregators for some task create a multi-party state channel, where the identifier is ID. The initial state of the channel is $ST_0 := \{st_{P_j}\}_{j \in [n]}$. After the channel is created, all participants $\{P_j\}_{j \in [n]}$ wait for the gradient shares sent from the training clients. The share aggregation is the same as the process above. After that, each participant $P_j$ obtains the aggregated gradient share $\langle g_j \rangle$. Then $P_j$ revokes PMSBFT.Pre$(\langle g_j \rangle)$ and obtains the aggregated signature $\Sigma_i$ with reconstructed gradient $g_j$, where $g \leftarrow \mathsf{SS.Reconstruct}(\{\langle g_j \rangle\}_{j \in [n]})$. After that, $P_j$ revokes PMSBFT.Com(msg.vote$_j^e$, msg.commit$_j^{e-1}$) with msg.commit$_j^{e-1}$ = (COMMIT, $e-1$, $g$) and msg.vote$_j^e$ = (VOTE, $e, \delta_j, \mathsf{msg}_i$), where $\delta_j = \mathsf{MulSig}(sk_j, \mathsf{msg}_i)$ and $\mathsf{msg}_i$ denotes the shares for another task $\tau_i$. The process of aggregated gradient reconstruction in the multi-party state channel with PMSBFT consensus is shown in Figure 2.

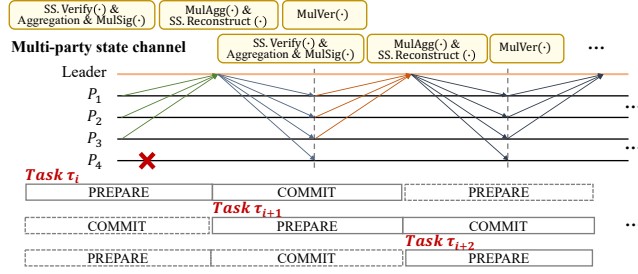

**Figure 2: The process of pipelined multi-signature BFT consensus** PMSBFT **protocol.**

**[Model updating and incentive distribution]** After the reconstruction, the participants in the channel obtain the aggregated

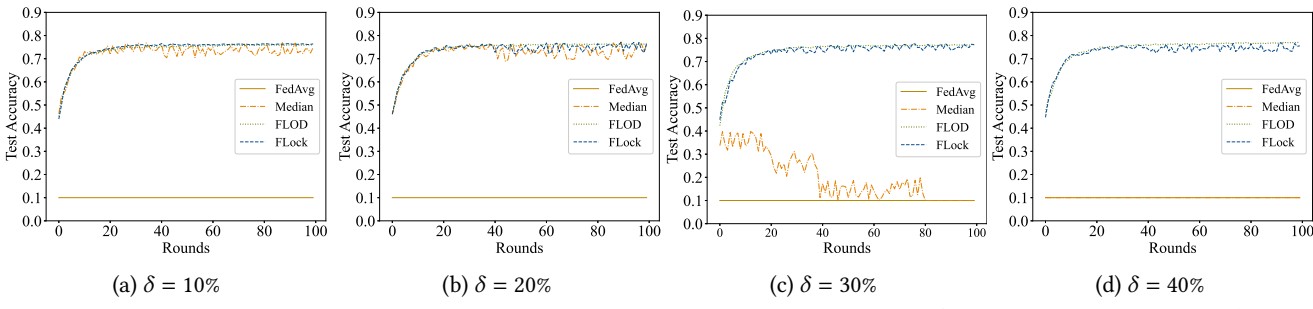

(a) $\delta = 10\%$      (b) $\delta = 20\%$      (c) $\delta = 30\%$      (d) $\delta = 40\%$

**Figure 3: Comparison of the test accuracy with 100 rounds of training iterations (FLOD adopts a root-dataset)**

gradient $g$. If the current gradient has reached the requirements of the task issuer $I_i$, the leader $\mathcal{L}$ in the channel will input it into the smart contract, and the smart contract will distribute rewards to the channel participants $\{P_j\}_{j \in [n]}$ and clients $\{C_l\}_{l \in [m]}$. The rewards for the channel participants are uniform, and the rewards for the clients are based on their scores $v^{(l)}$ in gradient aggregation.

## 5 Security Analysis

We capture the security of FLock in Theorem 1 and the concrete analysis and proofs are illustrated in Appendix D.

THEOREM 1. *FLock achieves poisoning robustness and cryptographic privacy on the top of the blockchain multi-party state channel, as long as i) there are less than 50% malicious clients, ii) the underlying protocols $\Pi_{\text{Bootstrap}}, \Pi_{\text{HM}}, \Pi_{\lambda\text{Score}}$, and $\Pi_{\text{WA}}$ are secure against static semi-honest adversaries in the Shamir's SS-hybrid model, and iii) the PMSBFT consensus is secure under the honest-majority assumption with $n \geq 3f + 1$, where $f$ is the maximum of corrupted aggregators and $n$ is the total number of aggregators.*

## 6 Evaluation

**Testbed Environment.** We conduct the experiments on a machine with Intel(R) Xeon(R) Sliver 4314 CPU @ 2.40GHz with 512GB RAM and Nvidia A100 with 40GB RAM. We simulate a WAN network setting: 400Mbps bandwidth and 4ms latency, with tc command. Our FL evaluation is based on FederatedScope [61] and we utilize LeNet [31] on dataset MNIST [18] and ResNet-20 [25] on dataset CIFAR-10 [30]. MPC protocols are developed on library hmmpc-public[2]. We finite field $\mathbb{F}_p$ with $p = 2^{61-1}$ with a fixed-point precision of 12 bits for MPC protocols. All of the instantiations related to blockchain are performed over the elliptic curve secp256k1, which is used on Bitcoin and Ethereum. To better demonstrate the compatibility of our work with Ethereum, we deploy the smart contracts on Ethereum with Solidity 0.8.0. Besides, we implement the PMSBFT consensus by using Go-1.22.1.

### 6.1 Poisoning Tolerance

We evaluate the accuracy of FLock under Gaussian attack [38], which is the most commonly used attack method. With the fraction of malicious clients $\delta$ to 10%, 20%, 30%, and 40%. We execute 100 rounds of training on ResNet over dataset CIFAR-10 [30] and compare the performance with FedAvg [41], Median[3] [64], and FLOD [19]. FedAvg is a vanilla FL scheme without any built-in defenses

against poisoning attacks, which performs optimally in the absence of such attacks. We measure the main task test accuracy, and experimental results are illustrated in Figure 3. We can see that as the training rounds increase, all methods gradually achieve their optimal test accuracy. However, as $\delta$ increases, the accuracy of Median and ours begin to fluctuate, while FLOD is relatively more stable, especially when $\delta = 30\%$ and $\delta = 40\%$. The performance of FLock is between FLOD and Median. However, when $\delta = 30\%$ and $\delta = 40\%$, even FLock is not as stable as FLOD, we still maintain comparable test accuracy. Besides, it is important to claim again that we do not require a clean root-dataset on the server side, while FLOD relies on this assumption. Besides, we compare the best test accuracy of these works on LeNet over dataset MNIST [18], the results are shown in Table 6. (c.f., Appendix C).

### 6.2 Secure Aggregation Efficiency

We evaluate the communication overhead (MB) and time cost (s) of $\Pi_{\text{Bootstrap}}, \Pi_{\text{HM}}, \Pi_{\lambda\text{Score}}$, and $\Pi_{\text{WA}}$. We set the number of aggregators to 7, 13, 19, and 25, where the performance of 7 and 25 aggregators is shown in Table 2 and the rest of the results are shown in the Appendix C. As mere prior works can support so many aggregators, for example, FLOD focuses on 2PC and cannot easily extended to more than 2 aggregators, we only measure our efficiency in experiments. From Table 2, we can see that: i) $\Pi_{\text{Boostrap}}$ and $\Pi_{\text{HM}}$ account for most of the communication and time costs for all experiments. This is expected since their cost is dependent on the size of the very large gradient vectors. ii) The cost of $\Pi_{\text{Boostrap}}$ and $\Pi_{\text{WA}}$ is only proportional to the size of gradient vectors but independent of the number of clients. This is because we only need to process one gradient vector. The cost of $\Pi_{\lambda\text{Score}}$ is the least as we only need to compute one score (*a.k.a.*, scalar) for each gradient. iii) The cost of $\Pi_{\text{Boostrap}}$ is mainly determined by computing $\langle g^{(s)} \rangle$. Though the size $\langle g^{(s)} \rangle$ is independent of the client number, it requires more costs than $\Pi_{\text{WA}}$ since LT is much more expensive than secure multiplication. Moreover, we can see similar efficiency observations for the results in Table 5 (c.f., Appendix C). As shown in Table 3, we also compare the aggregation overhead with FLOD[4] on ResNet, where FLOD is on ResNet-18 and FLock is on ResNet-20, which is heavier. Although our model has large parameters, our communication and run-time are much better than FLOD, confirming the efficiency of FLock's aggregation. Taking the setting with 100 clients and 7 aggregators as an example,

---

[2]https://github.com/f7ed/hmmpc-public
[3]Median is applied after the binarization process.

[4]Although FLOD is fixed to 2 aggregators, we compare FLock to it since it is the most relevant existing work to ours.

**Table 2: Aggregation efficiency with communication cost (MB) and run time (s) of Lenet (62K) and Resnet (273K)**

| # Agg. | Model | # Client | $\Pi_{Boostrap}$ | | $\Pi_{HM}$ | | $\Pi_{\lambda Score}$ | | $\Pi_{WA}$ | |
|---|---|---|---|---|---|---|---|---|---|---|
| | | | Comm. | Run-time | Comm. | Run-time | Comm. | Run-time | Comm. | Run-time |
| 7 | Lenet | 10 | 100.546 | 2.070 | 23.3823 | 0.401 | 0.016 | 0.015 | 2.338 | 0.081 |
| | | 50 | 100.548 | 2.072 | 116.914 | 1.857 | 0.081 | 0.015 | 2.338 | 0.081 |
| | | 100 | 100.550 | 2.074 | 233.828 | 3.504 | 0.162 | 0.015 | 2.338 | 0.081 |
| | Resnet | 10 | 442.731 | 9.144 | 102.96 | 1.510 | 0.016 | 0.015 | 10.296 | 0.027 |
| | | 50 | 442.730 | 9.162 | 514.8 | 18.430 | 0.081 | 0.015 | 10.296 | 0.027 |
| | | 100 | 442.732 | 9.174 | 1029.6 | 33.689 | 0.162 | 0.015 | 10.296 | 0.027 |
| 25 | Lenet | 10 | 401.885 | 8.505 | 90.470 | 2.433 | 0.065 | 0.030 | 9.047 | 0.248 |
| | | 50 | 401.886 | 8.508 | 452.352 | 10.132 | 0.324 | 0.452 | 9.047 | 0.248 |
| | | 100 | 401.894 | 8.518 | 904.704 | 21.109 | 0.648 | 0.495 | 9.047 | 0.248 |
| | Resnet | 10 | 1769.566 | 39.027 | 398.362 | 8.552 | 0.065 | 0.030 | 39.836 | 1.030 |
| | | 50 | 1769.567 | 39.277 | 1991.810 | 41.038 | 0.324 | 0.452 | 39.836 | 1.030 |
| | | 100 | 1769.575 | 39.352 | 3983.620 | 78.805 | 0.648 | 0.495 | 39.836 | 1.030 |

FLock reduces the communication costs by 76.26× and is around 54.16× faster compared to FLOD.

**Table 3: Comparison of the aggregation efficiency with FLOD in terms of communication cost (MB) and time (s) on ResNet**

| # Clients | Comm. | | | Run-time | | |
|---|---|---|---|---|---|---|
| | FLOD. On. | FLOD. Off. | FLock | FLOD. On. | FLOD. Off. | FLock |
| 10 | 403.0 | 11,745.280 | 556.004 | 43.170 | 192.040 | 10.696 |
| 50 | 1,982.610 | 58,030.080 | 967.970 | 202.760 | 969 | 27.634 |
| 100 | 3,957.650 | 109,117.440 | 1482.790 | 406.160 | 1917.430 | 42.905 |

## 6.3 On-chain and Off-chain Performance

We evaluate the on-chain and off-chain overhead involved in FLock, where on-chain cost is reflected by gas, and off-chain cost is reflected by time. In Ethereum, gas consumption reflects the complexity of the computation in the smart contracts. We set the gas price as 6 Gwei (as of Oct. 2024) and the exchange rate as 2315.8 USD per Ether. It can be seen from Table 4 that we increased the number of aggregators in the multi-party aggregation and then tested the time consumption of PMSBFT within the off-chain channel and the on-chain gas consumption of the multi-party state channel. The gas consumption includes the sum of the channel create and close. From the results, channel create and close cost a total of 227.8k gas ($4.138) when the number of participants is 7. It costs a total of 558.9k gas ($7.765) for a 37-party state channel. This price will fluctuate with the exchange rate of Ethereum.

**Table 4: Overhead of PMSBFT consensus (off-chain) within the channel and multi-party state channels (on-chain)**

| # of aggregators | | 7 | 13 | 19 | 25 | 31 | 37 |
|---|---|---|---|---|---|---|---|
| Off-chain (s) | | 0.620 | 1.227 | 4.685 | 6.249 | 8.291 | 13.370 |
| On-chain | Gas | 227815 | 254517 | 450019 | 486309 | 522600 | 558904 |
| | Ether/$10^{-3}$ | 1.367 | 1.527 | 2.700 | 2.918 | 3.136 | 3.353 |
| | USD | 4.138 | 3.536 | 6.253 | 6.758 | 7.262 | 7.765 |

## 7 Related Work

There are lots of works on privacy-preserving FL, many of which have achieved decentralization and can be combined with blockchain, and many works focus on secure aggregation. In Table 1, we compare some existing works from multiple dimensions.

Currently, many blockchain-based FL works have been proposed to achieve decentralization. However, most works have designed specialized consensus algorithms [2, 4, 15, 27, 33, 48, 51, 57, 66], which are only suitable on permissioned chains and are not compatible with existing permissionless cryptocurrency platforms, such as Bitcoin and Ethereum. Some works utilize a single server for gradient aggregation [42, 47, 58], which still has problems in centralization including a single point of failure. Besides, many studies have ignored the privacy of models and data [16, 22, 44, 54, 56], leading to potential security risks. First, there is a risk of leakage of private datasets for training, which makes it possible for the aggregator to reversely infer sensitive information of the original training data by analyzing intermediate results or gradient information [66]. Second, in a distributed environment, malicious clients can launch poisoning attacks to inject malicious or abnormal data into the system and disrupt the training process. This will not only affect the accuracy and performance of the global model after aggregation but may also cause model failure and even bring greater security threats. Some works use DP to protect models [20], but DP always has a direct negative impact on model accuracy.

Secure aggregation of FL is a widely studied technology that protects the privacy and security of data and models and resists inference attacks and poisoning attacks from aggregators and malicious clients. First, from the perspective of dataset privacy protection, there are many techniques have been adopted in existing works [40], including masking [5, 32, 37, 39, 57], *Additively Homomorphic Encryption* (AHE) [28, 52], *Fully Homomorphic Encryption* (FHE) [42], SS [29, 49], and some non-cryptographic methods [48, 51, 69]. Second, from the perspective of secure aggregation against malicious client poisoning, methods such as cosine similarity, Euclidean distance, Krum [8], root-dataset reference, and Hamming distance are commonly used. Third, in terms of the number of aggregators, most works are still aimed at single or two aggregators, and there is still less work on multi-party aggregation.

## 8 Conclusion

In this work, we propose FLock, a robust and privacy-preserving FL framework based on practical blockchain state channels. First, FLock achieves robustness against poisoning attacks from malicious clients through the proposed lightweight MPC-friendly robust aggregation method with quantization and Hamming distance optimizations. Second, FLock achieves privacy protection and resists inference attacks through our proposed multi-party secure aggregation taking advantage of Shamir's SS. Third, FLock is compatible with practical blockchain platforms, such as Ethereum, through multi-party state channels with PMSBFT consensus. Furthermore, this work provides fair incentives according to the contributions of participants by smart contracts. Moreover, we analyze the security of FLock. Evaluation results demonstrate our efficiency and practicality.

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

## A Pipelined Multi-signature BFT Consensus

### A.1 $(n, t)$-aggregable multi-signatures

The aggregable signatures allow multiple independent signatures to be aggregated into a single one and support batch verification of the validity of all signatures. The aggregable signatures aim to reduce data storage and computing resources, especially in scenarios such as distributed systems and blockchains. There are five algorithms involved in the aggregable signature scheme, which is shown in Appendix A.1. We set the aggregable signature to be $(n, t)$-threshold and instantiate the $(n, t)$-aggregable multi-signature based on the BLS signature [10]. We illustrate the $(n, t)$-aggregable multi-signature in Figure 4.

---

**$(n, t)$-Aggregable Multi-Signature**

- MulSetup$(\lambda) \rightarrow pp$: // generate the required public parameters
  - Initialize a bilinear group $(q, \mathbb{G}_1, \mathbb{G}_2, \mathbb{G}_t, e, g_1, g_2) \leftarrow \mathcal{G}(\lambda)$.
  - Output the public parameter $pp = (q, \mathbb{G}_1, \mathbb{G}_2, \mathbb{G}_t, e, g_1, g_2)$.
- MulKeyGen$(pp) \rightarrow (pk_i, sk_i)$: // generate key pair for each participant
  - Pick a $sk_i \leftarrow \mathbb{Z}_q$ and compute $pk_i \leftarrow g_2^{sk_i}$, $\rho_i \leftarrow \mathsf{H}_1(pk_i)^{sk_i}$.
  - Output $(pk_i, sk_i)$ and a public key list $\bar{\mathsf{PK}} = \{pk_1, \ldots, pk_n\}$.
- MulSig$(pp, sk_i, m) \rightarrow (\delta_i, m)$: // generate a signature share for $P_i$
  - Compute $\delta_i \leftarrow \mathsf{H}_0(m)^{sk_i}$ and output $(\delta_i, m)$.
- MulAgg$(\bar{\mathsf{PK}}, m, \{\delta_j, pk_j\}_{|t|}) \rightarrow (\Sigma, cpk, \boldsymbol{v}, m)$: // aggregate more than $t$ valid signature shares into a single one, where $t$ is the threshold
  - Initialize the signature list $\bar{\mathsf{SIG}} \leftarrow \emptyset$.
  - Verify if $(pk_j \in \bar{\mathsf{PK}}) \wedge (e(\delta_j, g_2) = e(\mathsf{H}_0(m), pk_j))$ for received $\delta_j$.
  - If the equation holds, then add $(\delta_j, pk_j)$ into $\bar{\mathsf{SIG}}$.
  - If $|\bar{\mathsf{SIG}}| \geq t$, then compute:
    $\Sigma \leftarrow \prod_{\{\delta_j | (\delta_j, \cdot) \in \bar{\mathsf{SIG}}\}} \delta_j$, $cpk \leftarrow \prod_{\{pk_j | (\cdot, pk_j) \in \bar{\mathsf{SIG}}\}} pk_j$.
  - Initiate a vector $\boldsymbol{v}$, where $\boldsymbol{v}[i] = \{0, 1\}$.
  - Let $\boldsymbol{v}[j]$ indicate whether $pk_j$ is in $\bar{\mathsf{SIG}}$ and output $(\Sigma, cpk, \boldsymbol{v}, m)$.
- MulVer$(\bar{\mathsf{PK}}, \Sigma, cpk, \boldsymbol{v}, m) \rightarrow 0/1$: // verify the validity of an aggregation of many signature shares at one time
  - Verify whether $cpk = \prod_{\{j | \boldsymbol{v}[j] = 1\}} \bar{\mathsf{PK}}[j]$ holds.
  - If the equation holds, then verify if $e(\Sigma, g_2) = e(\mathsf{H}_0(m), cpk)$.
  - If the equation does not hold, output 0. Otherwise, it outputs 1.

---

**Figure 4: The $(n, t)$-aggregable multi-signature scheme**

### A.2 The concrete protocol

The concrete PMSBFT protocol is described as follows.

- PMSBFT.Setup$(1^\kappa)$
  - Initiate the signature list $\bar{\mathsf{SIG}} := \bot$, vote list QC $:= \bot$, and commit list $\mathsf{QC}_\Sigma := \bot$.
- PMSBFT.Pre(msg) // Prepare phase
  ▷ **As a leader $\mathcal{L}$:** //Members $\mathcal{P} = \{P_1, \ldots, P_n\}$
  - Check if the current epoch $e$ has ended.
  - If Valid$(\mathsf{req}_i) = 1$:
    (1) Construct $\mathsf{msg}_i := (\text{Proposal}, \mathsf{req}_i)$.
    (2) Broadcast $(\mathsf{msg}_i, e, \Sigma_{i-1}, \mathsf{msg}_{i-1})$ among $\mathcal{P}$.
    (3) Start the timer $\Delta$.
  - Else, discard $\mathsf{req}_i$.
  ▷ **As a non-leader member $P_j$** $(1 \leq j \leq n)$
  - If received $(\mathsf{msg}_i, e, \Sigma_{i-1}, \mathsf{msg}_{i-1})$ such that MulVer$(\Sigma_{i-1}, \mathsf{msg}_{i-1}) = 1$, then commit $\mathsf{msg}_{i-1}$ and construct $\mathsf{msg.commit}_j^{e-1} := (\text{Commit}, e - 1, \mathsf{msg}_{i-1})$.
  - Else, discard $\Sigma_{i-1}$.
  - If Valid$(\mathsf{msg}_i) = 1$, generate $\delta_j \leftarrow \mathsf{MulSig}(sk_j, \mathsf{msg}_i)$.
  - Construct $\mathsf{msg.vote}_j^e := (\text{Vote}, e, \delta_j, \mathsf{msg}_i)$.
  - Send $(\mathsf{msg.vote}_j^e, \mathsf{msg.commit}_j^{e-1})$ to the leader $\mathcal{L}$.
- PMSBFT.Com$(\mathsf{msg.vote}_j^e, \mathsf{msg.commit}_j^{e-1})$ // Commit phase
  - If received $\mathsf{msg.vote}_j^e = (\text{Vote}, e, \delta_j, \mathsf{msg}_i)$ from $P_j$, s.t. Verify$(\delta_j, \mathsf{msg}_i) = 1$, set $\bar{\mathsf{SIG}} := \bar{\mathsf{SIG}} \cup \{\delta_j\}$, QC $:= \mathsf{QC} \cup \{\mathsf{msg.vote}_j^e\}$.
  - Else, discard $\mathsf{msg.vote}_j^e$.
  - If $|\mathsf{QC}| \geq 2n/3$ before $\Delta$:
    (1) Generate $(\Sigma_i, \mathsf{msg}_i) \leftarrow \mathsf{MulAgg}(\{\delta_j\}_{\delta_j \in \bar{\mathsf{SIG}}}, \mathsf{msg}_i)$.
    (2) Set the state of $\mathsf{msg}_i$ as $\mathsf{msg}_i.st := $ prepared.
  - If $(\Delta \text{ ends}) \wedge (|\mathsf{QC}| < 2n/3)$, reject and discard $\mathsf{msg.vote}_j^e$.
  - Upon $\mathsf{msg.commit}_j^{e-1} = (\text{Commit}, e - 1, \mathsf{msg}_{i-1})$ from $P_j$:
    (1) $\mathsf{QC}_\Sigma := \mathsf{QC}_\Sigma \cup \{\mathsf{msg.commit}_j^{e-1}\}$.
    (2) If $|\mathsf{QC}_\Sigma| \geq 2n/3$ before $\Delta$, set $\mathsf{msg}_{i-1}.st := $ committed.
    (3) Start the timer of the next epoch.

### A.3 Security definition of PMSBFT

The security of PMSBFT protocol is defined as follows.

**Table 5: Aggregation efficiency with communication cost (MB) and run time (s) of Lenet (62K) and Resnet (273K)**

| # Agg. | Model | # Client | $\Pi_{\text{Boostrap}}$ | | $\Pi_{\text{HM}}$ | | $\Pi_{\lambda\text{Score}}$ | | $\Pi_{\text{WA}}$ | | Total | |
|---|---|---|---|---|---|---|---|---|---|---|---|---|
| | | | Comm. | Run-time | Comm. | Run-time | Comm. | Run-time | Comm. | Run-time | Comm. | Run-time |
| 13 | Lenet | 10 | 200.997 | 4.024 | 45.785 | 0.743 | 0.032 | 0.018 | 4.578 | 0.105 | 251.392 | 4.890 |
| | | 50 | 200.998 | 4.026 | 228.923 | 3.844 | 0.162 | 0.024 | 4.578 | 0.105 | 434.661 | 7.999 |
| | | 100 | 201.001 | 4.028 | 457.846 | 7.537 | 0.324 | 0.030 | 4.578 | 0.105 | 663.750 | 11.700 |
| | Resnet | 10 | 885.027 | 18.377 | 201.6 | 3.281 | 0.032 | 0.018 | 20.16 | 0.664 | 1106.819 | 22.339 |
| | | 50 | 885.028 | 18.407 | 1008.0 | 15.196 | 0.162 | 0.024 | 20.16 | 0.664 | 1913.350 | 34.291 |
| | | 100 | 885.031 | 18.419 | 2016.0 | 31.080 | 0.324 | 0.030 | 20.16 | 0.664 | 2921.516 | 50.193 |
| 19 | Lenet | 10 | 301.442 | 6.131 | 68.135 | 1.211 | 0.049 | 0.025 | 6.813 | 0.228 | 376.438 | 7.596 |
| | | 50 | 301.442 | 6.136 | 340.673 | 6.170 | 0.243 | 0.028 | 6.813 | 0.228 | 649.172 | 12.562 |
| | | 100 | 301.448 | 6.340 | 681.347 | 12.439 | 0.486 | 0.031 | 6.813 | 0.228 | 990.095 | 19.038 |
| | Resnet | 10 | 1327.305 | 28.469 | 300.012 | 4.943 | 0.049 | 0.025 | 30.001 | 1.006 | 1657.366 | 34.444 |
| | | 50 | 1327.305 | 28.590 | 1500.06 | 24.572 | 0.243 | 0.028 | 30.001 | 1.006 | 2857.610 | 54.197 |
| | | 100 | 1327.311 | 28.613 | 3000.12 | 49.497 | 0.486 | 0.031 | 30.001 | 1.006 | 4357.918 | 79.148 |

---

**Algorithm 1** Task Issue

---

**Let** ADDR_CONTRACT be the address of smart contract
**Let** $Req_i$ be the statement of the task

---

**Init:** $\mathbb{C}_i \leftarrow \emptyset$
**Init:** $\exp_i$ for task expiry time
1: ▷ As task issuer $I_i$:
2: Set the expected accuracy $Acc_i$           ▷ $0 < Acc_i < 1$
3: Create constraint parameters $c_1, \ldots, c_m$
4: Let $\mathbb{C}_i := \mathbb{C}_i \cup \{c_1, \ldots, c_m\}$
5: Generate $\text{Tx}_i = (B_i, \text{ADDR\_CONTRACT})$     ▷ $B_i$ is the reward
6: Sign on $\text{Tx}_i$: $\sigma_i \leftarrow \text{Sign}_{sk_i}(\text{Tx}_i)$
7: Output $\tau_i = <Req_i, Acc_i, \mathbb{C}_i, B_i, \exp_i>$ and $\text{Tx}_i$
8: ▷ As a peer who executes smart contracts:
9: **Upon** receiving $\text{Tx}_i$ and $\tau_i$ **do**
10:    Parse $\tau_i = <Req_i, Acc_i, \mathbb{C}_i, B_i, \exp_i>$
11:    **if** $\text{SigVer}(pk_i, \text{Tx}_i, \sigma_i) = 1$ **then**
12:      **if** $\tau_i \notin \mathcal{T}$ **then**
13:        $\mathcal{T} := \mathcal{T} \cup \{\tau_i\}$
14:      **else** discard $\tau_i$ and stop

---

DEFINITION 1. *Let $\Pi_{\text{PMSBFT}}$ be a PMSBFT protocol under a partially synchronous network. Let $\delta_{\text{PMSBFT}} \leq \Delta$ be the actual network delay and $\mathsf{T}_{\text{PMSBFT}}$ be the upper bound of $\delta_{\text{PMSBFT}}$. $\Pi_{\text{PMSBFT}}$ is said to be secure if and only if the following properties hold.*

- *Safety: If any two honest participants $P_i$ and $P_j$ output the commit messages mag.commit$^e$ and mag.commit$'^e$ in same epoch $e$, then there must be mag.commit$^e$ = mag.commit$'^e$.*
- *Liveness: If there is a valid message req submitted by time $t$, then there must be a msg committed by all honest participants before time $t + \mathsf{T}_{\text{PMSBFT}}$.*

## B Task issue

The task issuer $I_i$ publishes the task on the chain. The format of a task is $\tau_i = <Req_i, Acc_i, \mathbb{C}_i, B_i, \exp_i>$, where $Req_i$ is the statement of requirements for the task, $Acc_i$ is the expectation for the model, *e.g.* accuracy, $\mathbb{C}_i$ is a list of constraints to the final model, such as specified algorithms, training data types, etc, $B_i$ is the bonus set by the task issuer for the task. The bonus will be automatically distributed by the smart contract to all contributor nodes participating in the training after the final model reaches the corresponding accuracy requirements. If the task result that satisfies $Acc_i$ has not been returned before $\exp_i$, the deposit will be returned to $I_i$. By setting the expiration time of the task, the funds of $I_i$ will be prevented from being permanently locked in the contract. When the block containing $\tau_i$ is uploaded on the chain, all other nodes in

**Table 6: Accuracy of LeNet over dataset MNIST.**

| $\delta$ | 10% | 20% | 30% | 40% |
|---|---|---|---|---|
| FedAvg [41] | 0.1135 | 0.1135 | 0.1135 | 0.1135 |
| Median [64] | 0.9895 | 0.9786 | 0.9490 | 0.8807 |
| FLOD [19] | 0.9930 | 0.9938 | 0.9935 | 0.9928 |
| FLock | 0.9998 | 0.9972 | 0.9494 | 0.9122 |

the network can see the task and decide whether to participate in training. All published tasks are added to a pending set $\mathcal{T}$.

## C More Evaluation Results

We provide the rest of the evaluation results here. We evaluate the test accuracy on MNIST and compare it with FedAvg, Median, and FLOD, which can be seen in Table 6. The aggregation performance of 13 aggregators and 19 aggregators can be found in Table 5.

## D Proof of Theorem 1

PROOF. Firstly, The poisoning robustness of FLock is supported by the bootstrapped gradients $\widehat{g}^{(s)}$ and the Hamming distance-based scoring with weighted aggregation.

- Since the gradients are quantized into binary values $\{-1, +1\}$, $\widehat{g}^{(s)}$ effectively represents the *element-wise median* of the client gradients $\widehat{g}^{(l)}_{l \in [m]}$. According to Theorems 1-3 of [64], median-based SGD ensures robustness in the presence of an honest majority under reasonable assumptions regarding the loss functions, gradients, and parameter space [64].
- According to the analysis and results of FLOD [19], the Hamming distance-based scoring and weighted aggregation method further strengthens resistance to common poisoning attacks.

Thus, our aggregation strategy ensures the poisoning robustness. Moreover, the poisoning robustness is empirically validated in our experiments (see § 6).

Secondly, we prove the security of $\Pi_{\text{Boostrap}}$, $\Pi_{\text{HM}}$, $\Pi_{\lambda\text{Score}}$, and $\Pi_{\text{WA}}$ described in § 4.3.1-§ 4.3.4.

- In $\Pi_{\text{Boostrap}}$, the security is fully based on the Shamir secret sharing scheme and LT, except that we remove the degree reduction with local computation of $\langle c^l \rangle^{2t}$ and $\langle \sigma^{(\ell)} \rangle^{2t}$, along with the opening of $\sigma^{(l)}$ in the **Test-then-Open** phase. Since we only change the order of these computations, the modification itself only involves some local computation and thus does not affect security.

- In $\Pi_{\mathsf{HM}}$, the security is fully based on the Shamir secret sharing scheme, except that we exploit a *Sum-then-DegReduce* technique to optimize the communication. In *Sum-then-DegReduce*, we change the order of addition and multiplication, which has no affect the security as well.
- In $\Pi_{\lambda\mathsf{Score}}$, the security is fully based on the Shamir secret sharing scheme since the protocol only involves some secret sharing multiplications and comparisons.
- In $\Pi_{\mathsf{WA}}$, the security is fully based on the Shamir secret sharing scheme, except that we make the sum of the scores $\mathcal{V}$ public and exploit a *Sum-then-DegReduce* technique. As mentioned before, *Sum-then-DegReduce* does not affect security. The sum of the scores $\mathcal{V}$ does not reveal the concrete score and gradients of each honest client. With the communication and computation improvements, it is a reasonable trade-off between efficiency and privacy.

Third, we prove the security of PMSBFT.

**THEOREM 2.** $\Pi_{\mathsf{PMSBFT}}$ *satisfies safety described in Definition 1 under a partially synchronous network with* $\delta_{\mathsf{PMSBFT}} \leq \Delta$ *and corrupted participants* $f < n/3$.

**PROOF.** We consider the normal process with a stable leader. For any epoch $e$, there should be only one valid aggregated signature, s.t. $\mathsf{MulVer}(\Sigma) = 1$. The unforgeability of the signature is guaranteed by the BLS signature scheme [10]. A valid aggregated signature $\Sigma$ is output means that at least $2f + 1$ participants constructed the voting message $\mathsf{msg.vote}^e$. If there are two different valid aggregated signatures $\Sigma$ and $\Sigma'$ with respect to $\mathsf{mag.commit}^e$ and $\mathsf{mag.commit'}^e$ within epoch $e$, then at least $2 \cdot (2f+1) - n = f+1$ participants voted for both $\Sigma$ and $\Sigma'$. Honest participants will only vote on one message in the same epoch, and the adversary $\mathcal{A}$ can corrupt at most $f$ participants. Thus, there will only be one valid commit message in the same epoch $e$, *i.e.* $\Sigma = \Sigma'$ and $\mathsf{mag.commit}^e = \mathsf{mag.commit'}^e$. □

**THEOREM 3.** $\Pi_{\mathsf{PMSBFT}}$ *satisfies liveness described in Definition 1 under a partially synchronous network with* $\delta_{\mathsf{PMSBFT}} \leq \Delta$ *and corrupted participants* $f < n/3$.

**PROOF.** Similarly, we consider the normal process with a stable leader. For any epoch $e$, a valid request message req sent by time $t$ will be processed within a time period. That is, there will be at least $2f + 1$ honest participants who vote for $\mathsf{msg} = (\textsc{Proposal}, \mathsf{req})$ and construct $\mathsf{vote.msg}^e = (\textsc{Vote}, e, \delta, \mathsf{msg})$ before the end of the current epoch $e$. Then the leader $\mathcal{L}$ constructs an aggregated signature $\Sigma \leftarrow \mathsf{MulAgg}(\{\delta\}_{|2f+1|})$ with $2f + 1$ valid signature shares. The participants receive the new proposal before $t + \delta_{\mathsf{PMSBFT}}$ and verify the validity of $\Sigma^{e-1}$ by $\mathsf{MulVer}(\Sigma^{e-1}) = 0/1$. Then the proposal of msg will be processed into the next phase since it has been approved by $2f + 1$ participants. Leader $\mathcal{L}$ receives at least $2f + 1$ votes at time $t + 2\delta_{\mathsf{PMSBFT}}$. Then $\mathcal{L}$ constructs the aggregate signature and changes the state of the message into the next phase. All participants receive the $\Sigma^e$ at time $t + 3\delta_{\mathsf{PMSBFT}}$. Then they verify the aggregated signature and construct the commit messages $\mathsf{msg.commit}^e$. Leader $\mathcal{L}$ receives $\mathsf{msg.commit}^e$ from at least $2f + 1$ participants at time $t + 4\delta_{\mathsf{PMSBFT}}$. Finally, the state of message msg will be set to $\mathsf{msg}.st := committed$ at time $t + 5\delta_{\mathsf{PMSBFT}}$.

Therefore, a valid message req submitted by time $t$ must be committed by all honest participants before time $t + \mathsf{T}_{\mathsf{PMSBFT}}$ with $\mathsf{T}_{\mathsf{PMSBFT}} = 5\delta_{\mathsf{PMSBFT}}$. □

Thus, we complete the security analysis of FLock. □

