# OpenReview forum: "FLock: Robust and Privacy-Preserving Federated Learning based on Practical Blockchain State Channels"
_ACM.org/TheWebConf/2025/Conference — WWW 2025 Oral_

### Official Review · Reviewer_UMZm · 2024-11-27

**Novelty:** 5
**Technical Quality:** 5

**Review:**

In Federated Learning (FL), it is difficult to achieve model robustness, data privacy, and accuracy simultaneously. The paper introduces FLock—the robust privacy-preserving FL that is compatible with blockchain (with low cost). The model defends against the poisoning attack of up to 50% of malicious clients (e.g., through quantization, hamming distance), and protects data privacy from aggregators (e.g., through Shamir secret sharing-based MPC). FLock also uses an off-chain state channel to achieve model immutability and distribute rewards on smart contracts. The authors finally evaluate FLock and show that even with 25 aggregators and 100 clients, the model completes ResNet in two minutes over a WAN.

Thank you for submitting your paper to WWW’25. I enjoyed reading the paper. While I have some concerns regarding the use of blockchain, I appreciate the authors' efforts to address multiple challenges, including privacy, robustness, and efficiency.

[Pros]
- The paper applies MPC to address the privacy concerns of aggregators inferring data from gradients.
- The paper implements the protocol in smart contracts and shows that the model is compatible with the existing blockchain such as Ethereum (through the off-chain state channel).
- The paper performs experiments to assess the aggregation latency.

[Cons]
- The practical applications of the model remain unclear.
- The use of blockchain can be better motivated.
- The reward distribution mechanism is unclear.
- The paper can benefit more from highlighting the difference from the existing methods (e.g., FLOD) and why FLock performs better than others.

**Questions:**

Can you provide a real-world example of what FLock is particularly useful for (over other FL methods)? It would be helpful to imagine what the practical application would be.

While the paper mentions the immutability of blockchain as an advantage, what does the model exactly try to achieve? Why is it necessary to record the weight on a chain? Is it for correctly distributing rewards for help training?

In Sec 3.2 threat model, did you set the ratio of an adversary to one-third for aggregators? The paper also has an assumption about 50% of malicious clients, so it might be helpful to clearly differentiate those two.

In Sec 4.2, how is lambda determined?

In Fig 2, what does the failure of P_4 mean in this context? The main text does not seem to mention this.

Why does FLock manage to significantly reduce the runtime costs than FLOD? How is the implementation of the aggregation method different? Some discussions would be helpful.

How does the contract distribute rewards to participants? Do the malicious actors get the reward even if they poison the model?

Can you publicly open-source the smart contract that you implement?

[Comments]

In Fig 1, the blockchain on the top is separated from the rest of the figure, so it is hard to tell what it does on its own.

It might be helpful to clarify the incentive for attackers to poison the model (i.e., send false gradients) and the consequence of poisoning.
The paper could benefit from referencing and citing some related works on this as well.

The techniques used in the training seem reasonable (e.g., boolean conversion).

In Sec 6.1, some explanation about the “Median” and “FLOD” methods would be helpful (even just adding pointers to Table 1 would be nice).

**Reviewer Confidence:**

3: The reviewer is confident but not certain that the evaluation is correct

**Scope:**

4: The work is relevant to the Web and to the track, and is of broad interest to the community

---

### Official Review · Reviewer_eoRp · 2024-11-28

**Novelty:** 4
**Technical Quality:** 5

**Review:**

1. In this paper, the author proposes FLock, a new scheme combining blockchain state channel and federated learning, which effectively solves the problems of data privacy, anti-neutrality and high model accuracy in federated learning.
2. FLock is flatly combined with mainstream blockchain to support low-cost decentralized federated learning and can achieve fair incentive mechanism through smart contracts.
3. The lightweight, multi-party computation-friendly aggregation method proposed in the paper dramatically reduces the communication and computation overhead through Shamir secret sharing protocol with optimized Hamming distance computation.

**Questions:**

1. Although the paper emphasizes the practical compatibility of FLock, it lacks a more detailed discussion of the impact of different sizes, diverse datasets, or dynamic network environments.
2. This thesis only assumes a “static semi-honest” attacker, and does not consider the scenario where a malicious participant changes its strategy during operation, or more advanced side-channel attacks, so the authors are requested to provide corresponding explanations or additional clarifications.
3. Incentive mechanism risk in decentralized environment: Although this thesis designs a contribution-based incentive mechanism, it does not fully consider the case of participants colluding or maliciously competing to obtain unfair rewards.

**Reviewer Confidence:**

2: The reviewer is willing to defend the evaluation, but it is likely that the reviewer did not understand parts of the paper

**Scope:**

4: The work is relevant to the Web and to the track, and is of broad interest to the community

---

### Official Review · Reviewer_UTN1 · 2024-11-28

**Novelty:** 6
**Technical Quality:** 7

**Review:**

This paper investigates the federated learning (FL) framework with security considerations. Specifically on poisoning robustness and data privacy and focuses on the practical compatibility to existing blockchain platforms. In addition, this paper introduces FLock, a robust and privacy-preserving FL framework based on practical blockchain state channels. This framework gives a practical and secure solution for decentralized FL, which shows strong potential and novelty.

Pros:
(1)	The secure aggregation method used in FLock achieves lightweight and robustness. One notable strength of this work is that it does not rely on a root-dataset for secure aggregation, which is a significant advantage.
(2)	Achieving low-overhead compatibility with blockchains through multi-party state channels is a very novel approach and is indeed superior to existing blockchain-based FL solutions in terms of practicality.
(3)	It is a very clever approach to perform secure aggregation through the idea of secret sharing and protect the privacy of data from poisoning attacks, because there is no need to sacrifice accuracy by adding noise to protect data.
(4)	The evaluation results demonstrate the claimed properties.

Cons:
(1)	The state channel seems to be an important component of FLock, but it is rarely described in the paper.
(2)	Misleading or unclear parts in the comparison table.

Detailed Comments:
1.	This paper proposes a novel and elegant privacy-preserving federated learning framework that achieves secure aggregation, resists poisoning attacks, and is practically compatible with existing blockchains. The framework uses multi-party state channels to improve practical compatibility with blockchains, but the description of multi-party state channels is almost only seen in 2.3.1. It would be better to add corresponding steps for channel create, update, and close in the protocol description or in the formal protocol description.
2.	The comparison table in the paper are well-constructed and provides valuable insights. However, some parts lack clarity. For instance, the methods used in existing works to achieve “Poisoning Robustness” are not clearly distinguished. Specifically, what is the difference between “median” and “Median-based testing”?
3.	Finally, some typos in the paper should be corrected. For instance, in 4.2, should the sentence before formula (3) be "$P$ computes $\textbf{g}_{k}^{(s)}$ to bootstrap trust in honest-majority"?

**Questions:**

1.	What content is stored and made public on-chain? What processes and data are handled off-chain? Additionally, how do the on-chain and off-chain components interact with each other to achieve the framework’s goals?
2.	The paper utilizes smart contracts for task issue and incentive distribution. Who is responsible for deploying the smart contracts in the proposed framework?
3.	In FLock, all aggregators participating in the aggregation of a specific model create a multi-party state channel to perform secure aggregation off-chain. The incentives for the aggregators are then evenly distributed among the participants of the channel. However, could there be a scenario where an aggregator node participates in multiple channels but does not perform any aggregation operations, yet still receives the corresponding rewards?

**Reviewer Confidence:**

4: The reviewer is certain that the evaluation is correct and very familiar with the relevant literature

**Scope:**

4: The work is relevant to the Web and to the track, and is of broad interest to the community

---

### Official Review · Reviewer_26Cx · 2024-12-01

**Novelty:** 5
**Technical Quality:** 5

**Review:**

The paper introduces FLock, a federated learning (FL) framework designed to address challenges in privacy, security, and decentralization through the integration of blockchain state channels. By leveraging blockchain state channels, it addresses key limitations in existing FL systems, offering robust defenses against poisoning attacks and privacy issues.

+ Pros

-The first framework to utilize off-chain state channels, enabling low-cost and scalable FL with fair incentives.
-Introduces a lightweight, MPC-friendly aggregation method based on quantization, Hamming distance, and median-based optimization, achieving resilience against up to 50% malicious clients without relying on a root dataset.
-Demonstrates compatibility with widely adopted blockchain platforms like Ethereum, showcasing real-world applicability through practical deployment experiments.

- Cons

-Large communication and computational overhead could become a bottleneck in larger-scale settings.
-The fairness of incentive mechanisms relies on accurate contribution scoring, which may not be reliable in scenarios with data imbalance or poorly performing clients.
-The paper does not address vertical federated learning (FL), limiting its applicability to scenarios where data is distributed across different features.

**Questions:**

Q1. Can FLock be applicable to vertical FL where participants hold different parts of a dataset?
Q2.While Table 1 offers a high-level comparison, a more detailed discussion of how FLock compares to state-of-the-art approaches in terms of computational overhead, scalability, and robustness would strengthen the paper.
Q3.It would be beneficial to provide a detailed breakdown of communication and computational overhead for various steps in the protocol.
Q4.To show the scalability, it would be beneficial to provide an asymptotic complexity with respect to the number of aggregators and clients.

**Reviewer Confidence:**

2: The reviewer is willing to defend the evaluation, but it is likely that the reviewer did not understand parts of the paper

**Scope:**

4: The work is relevant to the Web and to the track, and is of broad interest to the community